# EFFICIENT CONTINUOUS VIDEO FLOW MODEL FOR VIDEO PREDICTION

## ABSTRACT

Multi-step prediction models, such as diffusion and rectified flow models, have emerged as state-of-the-art solutions for generation tasks. However, these models exhibit higher latency in sampling new frames compared to single-step methods. This latency issue becomes a significant bottleneck when adapting such methods for video prediction tasks, given that a typical 60-second video comprises approximately 1.5K frames. In this paper, we propose a novel approach to modeling the multi-step process, aimed at alleviating latency constraints and facilitating the adaptation of such processes for video prediction tasks. Our approach not only reduces the number of sample steps required to predict the next frame but also minimizes computational demands by reducing the model size to one-third of the original size. We evaluate our method on standard video prediction datasets, including KTH, BAIR action robot, Human3.6M and UCF101, demonstrating its efficacy in achieving state-of-the-art performance on these benchmarks.

## 1 INTRODUCTION

Videos serve as digital representations of the continuous real world. However, due to inherent camera limitations, particularly fixed framerate constraints, these continuous signals are discretized in the temporal domain when captured. This temporal discretization, while often overlooked in generative video modeling, presents significant challenges. Current approaches, such as diffusion-based models, generally focus on generating the next frame in a sequence based on a given context frame, neglecting intermediate moments between frames (e.g., predicting frames at $T + 0.5$ or $T + 0.25$ given a context frame at $T$) as represented by Figure 1. In this work, we aim to address the limitations imposed by such discretization during the generative modeling of videos.

Diffusion models have emerged as a state-of-the-art technique for video generation, but they face computational challenges, particularly in inference time. High fidelity inferences from diffusion models require hundreds or even thousands of sampling steps, which may be feasible for single-image generation but becomes prohibitive for video generation, where thousands of frames must be generated for even a short video. The naive adaptation of diffusion models for video tasks results in significant computational bottlenecks during inference, especially in production settings. To address this issue, we propose a method that reduces the need for extensive sampling by starting the process from previous context frames rather than from an analytical distribution. This continuous modeling approach not only reduces the number of sampling steps but also enhances fidelity and reduces the model's parameter count.

Starting with two consecutive frames, we pass them through an encoder network to obtain their latent embeddings. We then interpolate between these endpoints using a noise schedule that applies zero noise at the boundaries, thus ensuring the existence of $p(\mathbf{x}_t)$ at all points. This continuous framework, builds on existing diffusion models for images while extending their applicability to videos.

In summary, our contributions are as follows:

- We introduce a novel approach for representing videos as multi-dimensional continuous processes in latent space.
- We empirically demonstrate that our model requires fewer sampling steps, reducing inference-time computational overhead without compromising result fidelity.

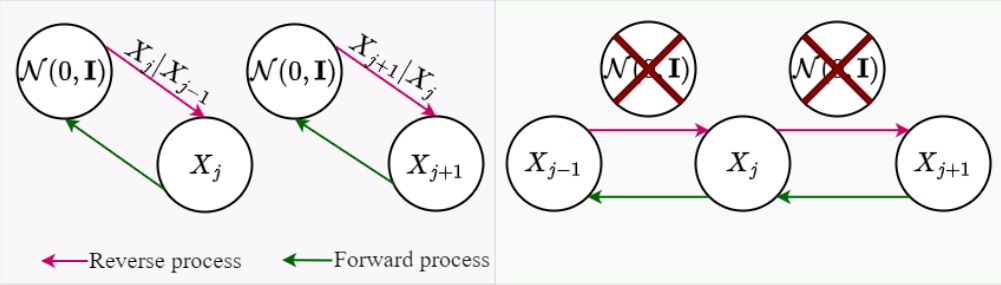

Figure 1: Fig. (a) represents a naive adaptation of the diffusion model for the video prediction task. Here, the sampling process always starts from a Gaussian distribution, and sampling steps are taken in the direction of conditional distribution given by $X_{j+1}|X_j$. Here, $X_{j+1}$ denotes frame at time $j+1$. In contrast, Fig. (b) introduces our Continuous Video Flow (CVF) approach, which reimagines the problem by treating video not as a discrete sequence of frames but as a continuously evolving process. Instead of starting from a static Gaussian distribution for each sampling step, CVF models the underlying dynamics of the entire video, learning to predict changes smoothly over time. This continuous framework allows the model to better capture temporal coherence and evolution, leading to more accurate and fluid video predictions.

- By reducing the memory footprint of frames in latent space, our model can predict future frames over a longer context sequence.

- Our approach requires fewer parameters compared to state-of-the-art video prediction models.

- We demonstrate state-of-the-art performance across several video prediction benchmarks, including KTH action recognition, BAIR robot push, Human3.6M, and UCF101 datasets.

## 2 RELATED WORKS

Predicting future frames from past observations is a fundamental task in video understanding, critical for domains requiring accurate modeling of multi-modal future outcomes, such as autonomous driving. Early approaches, such as those by Yuen et al. (Yuen & Torralba, 2010) and Walker et al. (Walker et al., 2014), tackled this problem by matching observed frames to predict future states, primarily relying on symbolic trajectories or directly retrieved future frames from datasets. While these methods laid important groundwork, their predictions were constrained by the deterministic nature of their models, limiting their ability to capture the inherent uncertainty of future frames.

The introduction of deep learning marked a significant advancement in this area. Srivastava et al. (Srivastava et al., 2015) pioneered the use of multi-layer LSTM networks, which focused on deterministic representation learning for video sequences. Building upon this, subsequent work explored more complex and stochastic methods. For instance, studies such as (Oliu et al., 2017; Cricri et al., 2016; Villegas et al., 2017; Elsayed et al., 2019; Villegas et al., 2019; Wang et al., 2019; Castrejón et al., 2019; Bodla et al., 2021) shifted towards incorporating stochastic processes to better model the uncertain nature of future frame predictions. This transition represents a growing recognition in the field: capturing uncertainty is essential for more accurate and robust future frame prediction.

Research in video prediction has advanced along two primary directions: implicit and explicit probabilistic models. Implicit models, particularly those rooted in Generative Adversarial Networks (GANs) (Goodfellow et al., 2014), have been widely explored but frequently encounter difficulties with training stability and mode collapse—where the model disproportionately focuses on a limited subset of data modes (Lee et al., 2018; Clark et al., 2019; Luc et al., 2020). These limitations have motivated increased interest in explicit probabilistic methods, which offer a more controlled approach to video prediction.

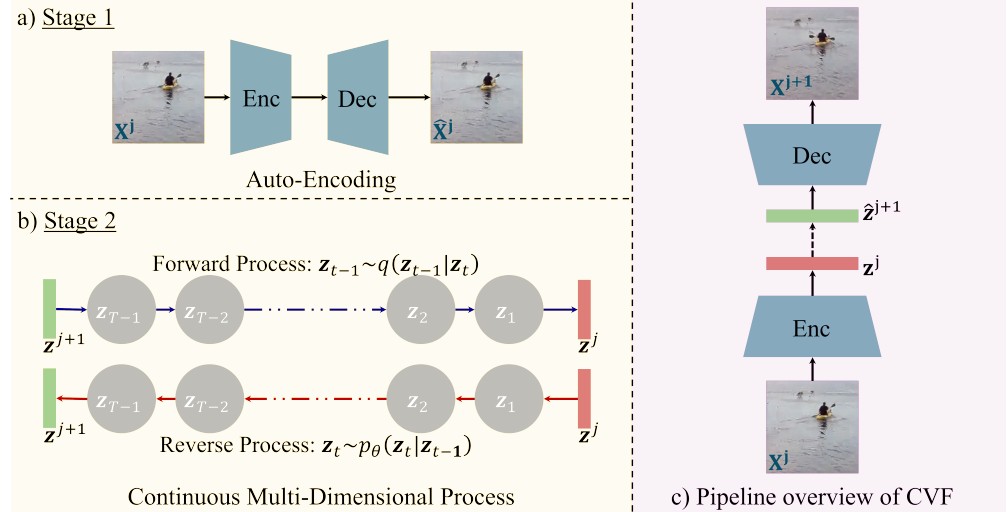

Figure 2: **Overview of the Continuous Video Flow (CVF) framework.** (a) Stage 1 depicts the auto-encoding process where input video frames $X_j$ are passed through an encoder (Enc) and decoder (Dec) to reconstruct $\hat{X}_j$. (b) Stage 2 illustrates the forward and reverse processes in the latent space. In the forward process, latent embeddings $z_T, \ldots, z_1$ are generated through a fixed process given by Eqn 2. The reverse process involves sampling $z_{j+1}$ from $z_j$ through the learned process $p_\theta$ for continuous video frame prediction. (c) The full pipeline of CVF, showing how frame $X_j$ is passed through the encoder to obtain latent embedding $z_j$ and $\hat{z}_{j+1}$, which is then used for decoding to obtain $\hat{X}_{j+1}$ frame.

Explicit methods cover a spectrum of techniques, such as Variational Autoencoders (VAEs) (Kingma & Welling, 2013), Gaussian processes, and diffusion models. VAE-based approaches to video prediction (Denton & Fergus, 2018; Castrejon et al., 2019; Lee et al., 2018) often produce results that average multiple potential outcomes, resulting in lower-quality predictions. Gaussian process-based models (Shrivastava & Shrivastava, 2021; Bhagat et al., 2020), while effective on small datasets, struggle with scalability due to the computational expense of matrix inversions required for training likelihood estimations. Attempts to circumvent this limitation usually result in reduced predictive accuracy.

Diffusion models (Voleti et al., 2022; Davtyan et al., 2023; Ho et al., 2022; Höppe et al., 2022) have emerged as a powerful alternative, providing high-quality samples and mitigating issues like mode collapse. These models leverage multi-step denoising, ensuring more consistent predictions. However, maintaining temporal coherence across frames requires additional constraints, such as temporal attention blocks, which can significantly increase computational demands.

An emerging method, the continuous video process (Shrivastava & Shrivastava, 2024), conceptualizes videos as continuously evolving data streams. This approach efficiently utilizes redundant information in video frames to reduce the time needed for inference. Despite its promise, this method relies on pixel-space blending between consecutive frames, leading to suboptimal redundancy exploitation. Our approach instead performs blending in the latent space, facilitating improved semantic interpolation between frames, as evidenced by previous work (Shrivastava & Shrivastava, 2024).

Finally, recent advancements such as InDI (Delbracio & Milanfar, 2023), rectified flow methods (Liu et al., 2022; Esser et al., 2024), and Cold Diffusion (Bansal et al., 2022) propose alternative diffusion-based strategies, though their focus has predominantly been on image generation and computational photography. In contrast, our approach extends these ideas to video prediction, achieving a balance between computational efficiency and temporal consistency.

## 3 METHOD

Videos contain significant redundancy at the pixel level, and directly interpolating between frames in pixel space often results in blurry intermediate frames. To address both the redundancy and

interpolation issues, we encode the frames into a latent space. This encoding offers two key benefits - First, **Compression**: Reduces irrelevant pixel information. Second, **Semantic Interpolation**: Ensures that interpolation in the latent space corresponds to more meaningful, semantic transitions between frames Shrivastava et al. (2023). In addition, encoding frames into the latent space allows for a larger context window when performing video prediction tasks.

### 3.1 Video Prediction Framework

Let $\mathcal{V} = \{\mathbf{x}^t\}_{t=1}^N$ be a video sequence where each frame $\mathbf{x}^j \in \mathbb{R}^{c \times h \times w}$ is a tensor representing the frame at timestep $j$. Our video prediction framework consists of two main stages:

**Stage 1: Encoding in Latent Space.** Each frame in the sequence is encoded into the latent space using a pre-trained autoencoder (Rombach et al., 2022). The encoder processes the video sequence $\mathcal{V}$ and produces a corresponding sequence of latent representations, frame by frame.

**Stage 2: Continuous Process in Latent Space.** After encoding the frames $\{\mathbf{x}^j, \mathbf{x}^{j+1}\}$ into the latent space, we model a continuous video process that transitions from the latent embedding of frame $\mathbf{x}^j$ to that of frame $\mathbf{x}^{j+1}$, following the framework proposed in Shrivastava & Shrivastava (2024). For clarity, $\mathbf{z}^j$ denotes the latent embedding of the frame at the discrete video timestep $j$, while $\mathbf{z}_t$ refers to the latent embedding at a continuous timestep $t$ during the interpolation between $\mathbf{x}^j$ and $\mathbf{x}^{j+1}$. In this context, $j$ indexes discrete video timesteps, whereas $t$ parameterizes the continuous video process. The interpolation between consecutive latent frames $\mathbf{z}^j$ and $\mathbf{z}^{j+1}$ is given by:

$$\mathbf{z}_t = (1-t)\mathbf{z}^j + t\mathbf{z}^{j+1} - \frac{t\log(t)}{\sqrt{2}}\boldsymbol{\epsilon} \tag{1}$$

where $\epsilon \sim \mathcal{N}(0, I)$ denotes white noise. This process ensures smooth transitions between frames in latent space. At $t = 0$, we retrieve the latent $\mathbf{z}^j$, and at $t = 1$, we retrieve the latent $\mathbf{z}^{j+1}$. For better understanding, refer Fig. 7

### 3.2 Forward Process

We utilize a forward process to transition from latent $\mathbf{z}^j$ to latent $\mathbf{z}^{j+1}$ over time $t$. The forward process starts at $t = T$ with latent $\mathbf{z}^{j+1}$ and moves to $t = 0$ for latent $\mathbf{z}^j$, as described by Eqn. 1 and further can be written as equation below:

$$\mathbf{z}_{t+\Delta t} = \mathbf{z}_t + (\mathbf{z}^{j+1} - \mathbf{z}^j)\Delta t - t\log(t)\boldsymbol{\epsilon} \tag{2}$$

From this, the forward process posterior is:

$$q(\mathbf{z}_{t+1}|\mathbf{z}_t, \mathbf{z}^j, \mathbf{z}^{j+1}) = \mathcal{N}(\mathbf{z}_{t+1}; \tilde{\mu}(\mathbf{z}_t, \mathbf{z}^j, \mathbf{z}^{j+1}), g^2(t)I) \tag{3}$$

where $g(t) = -t\log(t)$ and $\tilde{\mu}(\mathbf{z}_t, \mathbf{z}^j, \mathbf{z}^{j+1}) = \mathbf{z}_t + (\mathbf{z}^{j+1} - \mathbf{z}^j)$.

### 3.3 Likelihood and Variational Bound

We model the continuous video process in latent space through the following likelihood function:

$$p_\theta(\mathbf{z}_T) := \int p_\theta(\mathbf{z}_{0:T}) \, d\mathbf{z}_{0:T-1} \tag{4}$$

To train the model, we minimize the negative log-likelihood. The training objective involves minimizing the following variational bound:

$$\mathbb{E}[-\log p_\theta(\mathbf{z}_T)] \leq \mathbb{E}_q\left[-\log \frac{p_\theta(\mathbf{z}_{0:T})}{q(\mathbf{z}_{0:T-1}|\mathbf{z}_T)}\right] \tag{5}$$

which can be simplified following the paper (Shrivastava & Shrivastava, 2024):

$$L(\theta) = \sum_{t \geq 1} \text{KL}(q(\mathbf{z}_t|\mathbf{z}_{t-1}, \mathbf{z}^j, \mathbf{z}^{j+1})||p_\theta(\mathbf{z}_t|\mathbf{z}_{t-1}, \mathbf{z}^j)) \tag{6}$$

Here, the KL divergence compares the forward process posterior with the reverse process model.

**Algorithm 1** Sampling Algorithm

1: $\mathbf{z}^j \sim q_{\text{data}}(\mathbf{z}^j)$
2: $\mathbf{z}_0 = \text{Enc}(\mathbf{x})$
3: $d = \frac{1}{N}$,  Here $N$ denotes number of steps.
4: **for** $t = 1, \ldots, N$ **do**
5:   $\boldsymbol{\epsilon} \sim \mathcal{N}(\mathbf{0}, \mathbf{I}d)$ if $t > 1$, else $\boldsymbol{\epsilon} = \mathbf{0}$
6:   $\mathbf{z}_{t+1} = \mathbf{z}_t + (\hat{y}(\mathbf{z}_t, t) - \mathbf{z}^j)d - t\log(t)\boldsymbol{\epsilon}$
7: **end for**
8: **return** $\text{Dec}(\mathbf{z}_T)$

**Algorithm 2** Training of CVF model

1: **repeat**
2:   $\mathbf{x}, \mathbf{y} \sim q_{\text{data}}(\mathbf{x}, \mathbf{y})$
3:   $\mathbf{z}^j, \mathbf{z}^{j+1} \sim \text{Enc}(\mathbf{x}), \text{Enc}(\mathbf{y})$
4:   $t \sim \text{Uniform}(\{1, \ldots, T\})$
5:   $\boldsymbol{\epsilon} \sim \mathcal{N}(\mathbf{0}, \mathbf{I})$
6:   Take gradient descent step on
   $\nabla_\theta \frac{1}{2g^2(t)} \left\| \mathbf{z}^{j+1} - \mathbf{z}_\theta \left((1-t)\mathbf{z}^j + t\mathbf{z}^{j+1} - (t\log(t)/\sqrt{2})\boldsymbol{\epsilon}, t\right) \right\|^2$
7: **until** converged

## 3.4 REVERSE PROCESS

We assume a Markov chain structure for the reverse process, meaning that the current state depends only on the previous timestep:

$$p_\theta(\mathbf{z}_{0:T}) = p(\mathbf{z}_0) \prod_{t=1}^{T} p_\theta(\mathbf{z}_t | \mathbf{z}_{t-1}) \tag{7}$$

The reverse process is modeled as a Gaussian distribution:

$$p_\theta(\mathbf{z}_t | \mathbf{z}_{t-1}) = \mathcal{N}(\mathbf{z}_t; \boldsymbol{\mu}_\theta(\mathbf{z}_{t-1}, t-1), g^2(t)\mathbf{I}) \tag{8}$$

## 3.5 TRAINING OBJECTIVE

The training objective simplifies to:

$$L_{\text{simple}}(\theta) := \mathbb{E}_{t, \mathbf{z}_t} \left[ \frac{1}{2g^2(t)} \left\| \mathbf{z}^{j+1} - \mathbf{z}_\theta(\mathbf{z}_t, t) \right\|^2 \right] \tag{9}$$

where $g(t) = -t\log(t)$ controls the noise added at each timestep. We use the interpolation function in Eqn. 1 to model the intermediate frames during training.

**Final Loss Function.** The final training objective to optimize the video prediction model is given by:

$$\arg\min_\theta \mathbb{E}_{t, \mathbf{z}^j, \mathbf{z}^{j+1}} \left[ \frac{1}{2g^2(t)} \left\| \mathbf{z}^{j+1} - \mathbf{z}_\theta((1-t)\mathbf{z}^j + t\mathbf{z}^{j+1} + \frac{g(t)}{\sqrt{2}}\boldsymbol{\epsilon}, t) \right\|^2 \right] \tag{10}$$

The entire training and sampling pipeline is described in Algorithm 2 and Algorithm 1, and visualized in Figure 2.

## 4 EXPERIMENTS

The video prediction task is defined as predicting future frames from a given sequence of context frames. In this section, we empirically validate the performance of our proposed method, demonstrating its superior results in modeling video prediction tasks across diverse datasets.

### 4.1 DATASETS

We employ four standard benchmarks to evaluate the efficacy of our approach: KTH Action Recognition, BAIR Robot Pushing, Human3.6M, and UCF101. Each of these datasets presents unique challenges for video prediction, allowing for a comprehensive assessment of our method's robustness. Training protocols and architecture specifics are provided in the appendix.

**KTH Action Recognition Dataset:** This dataset (Schuldt et al., 2004) contains video sequences of 25 individuals performing six actions—walking, jogging, running, boxing, hand-waving, and

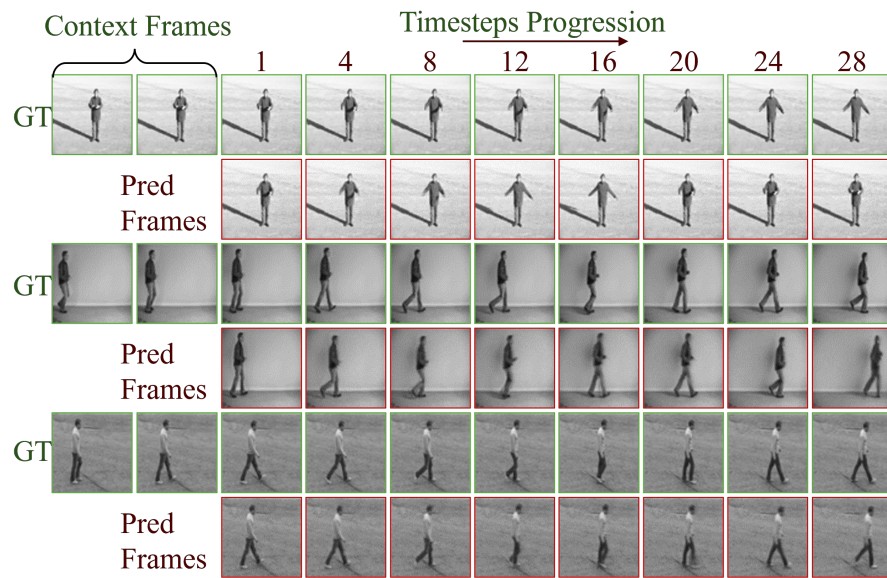

Figure 3: Figure represents qualitative results of our CVF model on the KTH dataset. The number of context frames used in the above setting is 4 for all three sequences. Every $4^{th}$ predicted future frame is shown in the figure.

Table 1: Video prediction results on KTH ($64 \times 64$), predicting 30 and 40 frames using models trained to predict $k$ frames at a time. All models condition on 10 past frames on 256 test videos.

| KTH [$10 \rightarrow$ #pred; trained on $k$] | $k$ | #pred | FVD↓ | PSNR↑ | SSIM↑ |
|---|---|---|---|---|---|
| SVG-LP (Denton & Fergus, 2018) | 10 | 30 | 377 | 28.1 | 0.844 |
| SAVP (Lee et al., 2018) | 10 | 30 | 374 | 26.5 | 0.756 |
| MCVD (Voleti et al., 2022) | 5 | 30 | 323 | 27.5 | 0.835 |
| SLAMP (Akan et al., 2021) | 10 | 30 | 228 | 29.4 | 0.865 |
| SRVP (Franceschi et al., 2020) | 10 | 30 | 222 | 29.7 | 0.870 |
| RIVER (Davtyan et al., 2023) | 10 | 30 | 180 | 30.4 | 0.86 |
| CVP (Shrivastava & Shrivastava, 2024) | 1 | 30 | 140.6 | 29.8 | 0.872 |
| **CVF (Ours)** | **1** | 30 | **108.6** | 30.6 | **0.891** |
| Struct-vRNN (Minderer et al., 2019) | 10 | 40 | 395.0 | 24.29 | 0.766 |
| SVG-LP (Denton & Fergus, 2018) | 10 | 40 | 157.9 | 23.91 | 0.800 |
| MCVD (Voleti et al., 2022) | 5 | 40 | 276.7 | 26.40 | 0.812 |
| SAVP-VAE (Lee et al., 2018) | 10 | 40 | 145.7 | 26.00 | 0.806 |
| Grid-keypoints (Gao et al., 2021) | 10 | 40 | 144.2 | 27.11 | 0.837 |
| RIVER (Davtyan et al., 2023) | 10 | 40 | 170.5 | 29.0 | 0.82 |
| CVP (Shrivastava & Shrivastava, 2024) | 1 | 40 | 120.1 | 29.2 | 0.841 |
| **CVF (Ours)** | **1** | 40 | **100.8** | **29.7** | **0.852** |

hand-clapping. The videos feature a uniform background with a single person performing the action in the foreground, offering relatively regular motion patterns. Each video frame is downsampled to a spatial resolution of $64 \times 64$ and is represented as a single-channel image.

**BAIR Robot Pushing Dataset:** The BAIR dataset (Ebert et al., 2017) captures a Sawyer robotic arm pushing various objects on a table. It includes different robotic actions, providing a dynamic and controlled environment. The resolution of the video frames is also $64 \times 64$.

**Human3.6M Dataset:** Human3.6M (Catalin Ionescu, 2011) features 10 subjects performing 15 different actions. Unlike other works, we exclude pose information from the prediction task, focusing solely on video frame data. The dataset offers regular foreground motion against a uniform background, with video frames in RGB format and downsampled to $64 \times 64$.

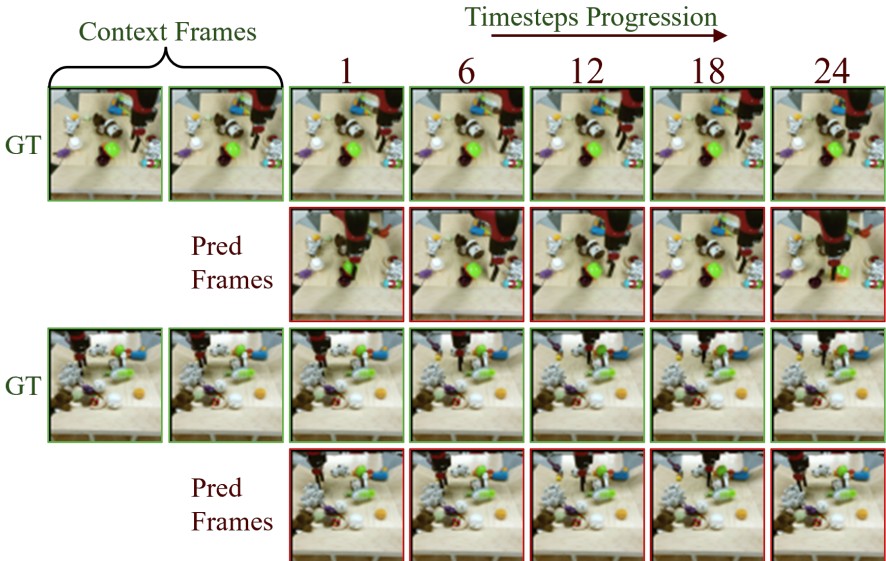

Figure 4: Figure represents qualitative results of our CVF model on the BAIR dataset. The number of context frames used in the above setting is two for both sequences. Every $6^{th}$ predicted future frame is shown in the figure.

**UCF101 Dataset:** UCF101 (Soomro et al., 2012) consists of 13,320 videos spanning 101 action classes. This dataset introduces a diverse range of backgrounds and actions. Each frame is resized from the original resolution of $320 \times 240$ to $128 \times 128$ using bicubic downsampling, retaining all three RGB channels.

Table 2: *BAIR* dataset evaluation. Video prediction results on BAIR ($64 \times 64$) conditioning on $p$ past frames and predicting $pred$ frames in the future, using models trained to predict $k$ frames at at time.The common way to compute the FVD is to compare $100 \times 256$ generated sequences to 256 randomly sampled test videos. Best results are marked in *bold*.

| **BAIR** ($64 \times 64$) | $p$ | $k$ | #pred | FVD↓ |
|---|---|---|---|---|
| LVT (Rakhimov et al., 2020) | 1 | 15 | 15 | 125.8 |
| DVD-GAN-FP (Clark et al., 2019) | 1 | 15 | 15 | 109.8 |
| TrIVD-GAN-FP (Luc et al., 2020) | 1 | 15 | 15 | 103.3 |
| VideoGPT (Yan et al., 2021) | 1 | 15 | 15 | 103.3 |
| CCVS (Le Moing et al., 2021) | 1 | 15 | 15 | 99.0 |
| FitVid (Babaeizadeh et al., 2021) | 1 | 15 | 15 | 93.6 |
| MCVD (Voleti et al., 2022) | 1 | 5 | 15 | 89.5 |
| NÜWA (Liang et al., 2022) | 1 | 15 | 15 | 86.9 |
| RaMViD (Höppe et al., 2022) | 1 | 15 | 15 | 84.2 |
| VDM (Ho et al., 2022) | 1 | 15 | 15 | 66.9 |
| RIVER (Davtyan et al., 2023) | 1 | 15 | 15 | 73.5 |
| CVP (Shrivastava & Shrivastava, 2024) | 1 | 1 | 15 | 70.1 |
| CVF (Ours) | 1 | 1 | 15 | 65.8 |
| DVG (Shrivastava & Shrivastava, 2021) | 2 | 14 | 14 | 120.0 |
| SAVP (Lee et al., 2018) | 2 | 14 | 14 | 116.4 |
| MCVD (Voleti et al., 2022) | 2 | 5 | 14 | 87.9 |
| CVP (Shrivastava & Shrivastava, 2024) | 2 | 1 | 14 | 65.1 |
| **CVF(Ours)** | 2 | **1** | 14 | **61.2** |
| SAVP (Lee et al., 2018) | 2 | 10 | 28 | 143.4 |
| Hier-vRNN (Castrejon et al., 2019) | 2 | 10 | 28 | 143.4 |
| MCVD (Voleti et al., 2022) | 2 | 5 | 28 | 118.4 |
| CVP (Shrivastava & Shrivastava, 2024) | 2 | 1 | 28 | 85.1 |
| **CVF (Ours)** | 2 | **1** | 28 | **78.5** |

Table 3: Comparison with baselines on number of parameters, sampling steps and sampling time required for BAIR robot push dataset.

| BAIR | # Params | # Sampling(Steps/Frame) | Time Taken(in hrs) |
|------|----------|-------------------------|--------------------|
| MCVD (Voleti et al., 2022) | 251M | 100 | 2 |
| RaMViD (Höppe et al., 2022) | 235M | 500 | 7.2 |
| CVP (Shrivastava & Shrivastava, 2024) | 118M | 25 | 0.45 |
| **CVF (Ours)** | **40M** | **5** | **0.112** |

## 4.2 METRICS

For performance evaluation, we use the Fréchet Video Distance (FVD) (Unterthiner et al., 2018) metric, which assesses both reconstruction quality and diversity in generated samples. The FVD computes the Fréchet distance between I3D embeddings of generated and real video samples. The I3D network is pretrained on the Kinetics-400 dataset to provide reliable embeddings for this metric.

## 5 SETUP AND RESULTS

In this section, we detail the experimental setup, comparing our method to existing baselines. We then present the performance of our approach across all datasets, analyzing both quantitative metrics and qualitative results.

**KTH Action Recognition Dataset:** For this dataset, we followed the baseline setup from (Shrivastava & Shrivastava, 2024), which uses the first 10 frames as context to predict the subsequent 30 or 40 future frames. A key distinction in our experiment is that, instead of utilizing all 10 context frames, we only use the last 4 frames as context in our CVF model, deliberately discarding the first 6 frames. This choice aligns with prior methodologies and allows a fair comparison. The results of our evaluation are summarized in Table 1.

From Table 1, it is evident that our model achieves superior performance while requiring fewer training frames. Unlike other approaches that rely on a larger number of frames for both context and future predictions (e.g., 10 context frames plus $k$ future frames), our model operates effectively with just 4 context frames and 1 future frame. Specifically, we predict the immediate next frame using 4 context frames and then autoregressively generate 30 or 40 future frames, depending on the evaluation setting. This efficiency stems from our model's continuous sequence processing capabilities, which avoid the need for explicit temporal attention mechanisms or artificial constraints.

The results, as presented in Table 1, confirm that our method delivers state-of-the-art performance relative to baseline models. Qualitative results of our CVF model on the KTH dataset are shown in Fig. 3.

**BAIR Robot Push Dataset:** The BAIR Robot Push dataset is known for its highly stochastic video sequences. In line with previous studies (Shrivastava & Shrivastava, 2024), we experimented with three different setups: 1) using one context frame to predict the next 15 frames, 2) using two context frames to predict 14 future frames, and 3) using two context frames to predict 28 future frames. The results for these settings are detailed in Table 2.

As shown in Table 2, there is a clear trend where increasing the number of predicted frames leads to a gradual decline in prediction quality. This performance drop is likely due to an increasing mismatch between the context frames and the predicted future frames. For instance, when using two context frames, denoted as $\mathbf{x}^{0:2}$, to predict a single future frame, the prediction block is represented as $\mathbf{z}^{1:3}$, aligning with the setup in Eqn.1. In the second scenario, where two frames are predicted, the future block extends to $\mathbf{x}^{2:4}$. This setup means that in the first condition, interpolation occurs between adjacent frames (i.e., from $\mathbf{z}^0 \to \mathbf{z}^1$ and $\mathbf{z}^1 \to \mathbf{z}^2$), while in the second condition, interpolation spans a larger gap (e.g., $\mathbf{z}^0 \to \mathbf{z}^2$ and $\mathbf{z}^1 \to \mathbf{z}^3$). This extended gap likely contributes to the observed decrease in predictive performance, particularly when $k = 2$ and $p = 2$.

As indicated in Table 2, our method consistently outperforms baseline models. Qualitative results of our CVF model on the BAIR dataset can be seen in Fig. 4.

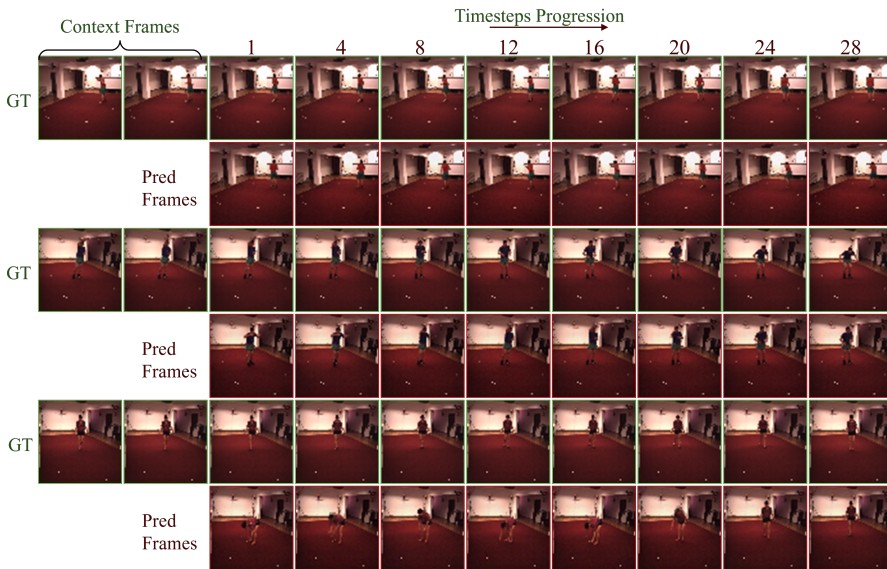

Figure 5: Figure represents qualitative results of our CVF model on the Human3.6M dataset. The number of context frames used in the above setting is 4 for all three sequences. Every $4^{th}$ predicted future frame is shown in the figure.

Table 4: Quantitative comparisons on the Human3.6M dataset. The best results under each metric are marked in bold.

| Human3.6M | p | k | #pred | FVD↓ |
|---|---|---|---|---|
| SVG-LP (Denton & Fergus, 2018) | 5 | 10 | 30 | 718 |
| Struct-VRNN (Minderer et al., 2019) | 5 | 10 | 30 | 523.4 |
| DVG (Shrivastava & Shrivastava, 2021) | 5 | 10 | 30 | 479.5 |
| SRVP (Franceschi et al., 2020) | 5 | 10 | 30 | 416.5 |
| Grid keypoint (Gao et al., 2021) | 8 | 8 | 30 | 166.1 |
| CVP (Shrivastava & Shrivastava, 2024) | 5 | 1 | 30 | 144.5 |
| **CVF(Ours)** | 5 | 1 | 30 | **120.2** |

Table 3 highlights the superior efficiency of the proposed CVF model compared to diffusion based baselines across key metrics. CVF has the fewest parameters and requires only 5 sampling steps per frame. This makes CVF highly efficient and practical for video prediction tasks, where speed and resource efficiency are critical.

**Human3.6M Dataset:** Like the KTH dataset, the Human3.6M dataset features actors performing distinct actions against a static background. However, Human3.6M provides a broader range of actions and includes three-channel RGB video frames, whereas KTH offers single-channel frames. For our evaluation, we adopted a setup similar to that used for KTH, providing 5 context frames and predicting the subsequent 30 future frames. The results of this evaluation are summarized in Table 4.

From Table 4, it is clear that our model requires significantly fewer training frames to achieve superior results. Specifically, it only uses 6 frames per block (5 context frames and 1 future frame), yet delivers performance that surpasses the baseline methods.

The results, as shown in Table 4, demonstrate that our approach outperforms existing models, establishing a new state-of-the-art on the Human3.6M dataset. Additionally, the qualitative results in Fig. 5 highlight our CVF model's ability to accurately capture and predict the diverse actions within the dataset, further showcasing its effectiveness.

**UCF101 Dataset:** The UCF101 dataset introduces a significantly higher level of complexity compared to the KTH and Human3.6M datasets, primarily due to its large variety of action categories, diverse backgrounds, and pronounced camera movements. For our frame-conditional generation task,

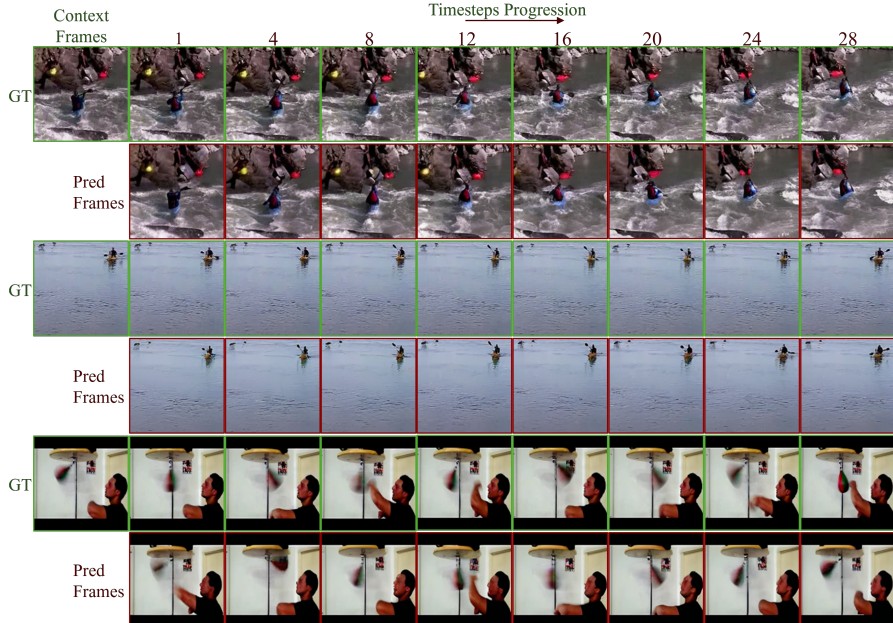

Figure 6: Figure represents qualitative results of our CVF model on the UCF dataset. The number of context frames used in the above setting is 5 for all three sequences. Every $4^{th}$ predicted future frame is shown in the figure.

Table 5: Video prediction results on UCF ($128 \times 128$), predicting 16 frames. All models are conditioned on 5 past frames.

| **UCF101** [$5 \rightarrow 16$] | $p$ | $k$ | #pred | FVD↓ |
|---|---|---|---|---|
| SVG-LP (Denton & Fergus, 2018) | 5 | 10 | 16 | 1248 |
| CCVS (Le Moing et al., 2021) | 5 | 16 | 16 | 409 |
| MCVD (Voleti et al., 2022) | 5 | 5 | 16 | 387 |
| RaMViD (Höppe et al., 2022) | 5 | 4 | 16 | 356 |
| CVP (Shrivastava & Shrivastava, 2024) | 5 | 1 | 16 | 245.2 |
| **CVF(Ours)** | 5 | 1 | 16 | **221.7** |

we strictly utilize information from the context frames, without leveraging any additional data, such as class labels, for the prediction task. Our evaluation setup mirrors that used for the Human3.6M dataset, where 5 context frames are provided, and the model is tasked with predicting the subsequent 16 frames. The results of this evaluation are presented in Table 5.

Upon analyzing Table 5, it is evident that our CVF model outperforms all baseline models, setting a new state-of-the-art benchmark for the UCF101 dataset. Furthermore, the qualitative results shown in Fig. 6 demonstrate the model's ability to effectively capture and predict the diverse actions present in this challenging dataset, even without the use of external information like class labels.

## 6 CONCLUSION

In this work, we introduced a novel model architecture designed to more effectively leverage video representations, marking a significant contribution to video prediction tasks. Through extensive experimental evaluation on diverse datasets, including KTH, BAIR, Human3.6M, and UCF101, our approach consistently demonstrated superior performance, setting new benchmarks in state-of-the-art video prediction.

A key strength of our model is its efficient utilization of parameters and significantly lower number of sampling steps during inference compared to existing methods. Notably, our model's ability to treat video as a continuous process eliminates the need for additional constraints, such as temporal attention mechanisms, which are often used to enforce temporal consistency. This highlights the

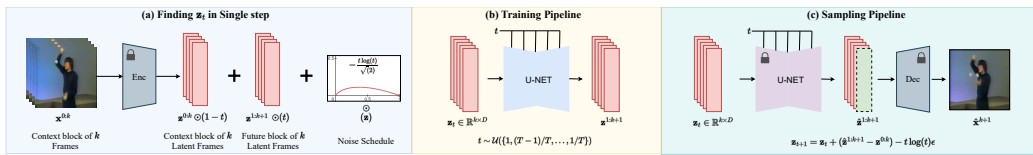

Figure 7: (a) Illustration of the single-step estimation process for $\mathbf{z}_t$, where a pre-trained Encoder encodes a block of $k$ frames, highlighting the computational methodology employed. (b) The training pipeline of the Continuous Video Process (CVP) model, where $\mathbf{z}_t$ and $t$ serve as inputs to the U-Net architecture (details in Appendix), producing the predicted output $\hat{\mathbf{z}}^{1:k+1}$. (c) Overview of the sampling pipeline used in our approach, demonstrating the sequential prediction of the next frame in the video sequence. Given context frames in latent space $\hat{\mathbf{z}}^{0:k}$, the predicted latent $\hat{\mathbf{z}}^{k+1}$ is decoded to generate the subsequent frame $\hat{\mathbf{x}}^{k+1}$.

model's inherent capacity to maintain temporal coherence naturally, streamlining the video prediction process while improving both efficiency and predictive accuracy.

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
