```
702   model = Gpt(
703       parameter_sizes=[z_dim*n],  # z_dim is latent space dimension
704       parameter_names=['weight'],
705       predict_xstart=True,
706       absolute_loss_conditioning=False,
707       chunk_size=64,  # cfg.transformer.chunk_size,
708       max_freq_log2=20,
709       num_frequencies=128,
710       n_embd=64,  # cfg.transformer.n_embd,
711       encoder_depth=2,  # cfg.transformer.encoder_depth,
712       decoder_depth=2,  # cfg.transformer.decoder_depth,
713       n_layer=768,  # cfg.transformer.n_layer,
714       n_head=16,  # cfg.transformer.n_head,
715       attn_pdrop=0.1,  # cfg.transformer.dropout_prob,
716       resid_pdrop=0.1,  # cfg.transformer.dropout_prob,
717       embd_pdrop=0.1  # cfg.transformer.dropout_prob
718   )
```

Figure 7: **GPT modified transformer for diffusion Peebles et al. (2022) in latent space:** The latent embedding dimension for KTH, BAIR and Human3.6M is kept at $64$ and $128$ for the UCF101 dataset. Additionally, we keep the number of timesteps $T$ as 100 given our compute resources. $n$ is the number of initial context frames based on which next frame is predicted,i.e., $\mathbf{z}^{0:n} \rightarrow \mathbf{z}^{1:n+1}$. Also, for KTH, Human3.6M, and BAIR datasets we used vgg-based autoencoders Shrivastava & Shrivastava (2021). For UCF101 we used pretrained autoencoder Rombach et al. (2022).

## A    TRAINING DETAILS

For the optimization of our model, we harnessed the compute of two Nvidia A6000 GPUs, each equipped with 48GB of memory, to train our CVF model effectively. We adopted a batch size of 64 and conducted training for a total of 500,000 iterations. To optimize the model parameters, we employed the AdamW optimizer. Additionally, we incorporated a cosine decay schedule for learning rate adjustment, with warm-up steps set at 10,000 iterations. The maximum learning rate (Max LR) utilized during training was 5e-5.

## B    LIMITATION

While our method demonstrates strong performance in video prediction, it is essential to acknowledge certain limitations that point toward avenues for future work.

First, a key limitation lies in computational efficiency. Although our approach requires fewer sampling steps compared to traditional diffusion-based models, generating each frame still demands a sequential process that can become a bottleneck when scaling to longer video sequences or real-time applications. Further optimization, particularly in reducing the number of sampling steps and computational overhead, remains an open challenge.

Second, our experiments were constrained by computational resources, utilizing only two A6000 GPUs. With access to more powerful hardware or distributed computing, there may be potential for significant gains in both model complexity and performance. We encourage future research to investigate the model's behavior on larger datasets and with more substantial computational resources, as these factors could reveal additional improvements in video prediction quality.