# OpenReview forum: "Efficient Continuous Video Flow Model for Video Prediction"
_ICLR.cc/2025/Conference — Submitted to ICLR 2025_

### Official Review · Reviewer_FoaZ · 2024-10-30

**Soundness:** 2
**Presentation:** 3
**Contribution:** 2
**Rating:** 5
**Confidence:** 4

**Summary:**

The paper addresses latency constraints in diffusion-based video prediction, which arise from the need for multi-step sampling. This work proposes a method that represents videos as multi-dimensional continuous processes in latent space, allowing for reduced sampling steps and more efficient prediction of future video frames. Experiments are conducted on four benchmark video prediction datasets.

**Strengths:**

1) Motivation: The need to reduce latency in video generation is well-motivated and thoroughly explained.
2) Technical Soundness: The proposed model is concise and technically sound.
3) Clarity of Writing: The paper is well-written, making it easy to follow and understand.

**Weaknesses:**

The reviewer thinks the motivation of this paper is good, however, the contribution of this paper is incremental.

Two main contributions are claimed in the paper,
1) Latent Video Representation: The paper proposes using a latent representation of videos/frames to reduce computational costs. However, leveraging latent visual representations to address computational efficiency is a recognized practice within the diffusion community. Prior work, such as PVDM [1] and Seer [2], has already demonstrated similar methods.
2) With the latent video representation, the second one is representing videos as multi-dimensional continuous processes. However, this seems to be a well-established framework used for this task. For example, the CVP [3], which is the previous SOTA compared in the paper, used this framework to generate the video futures.

[1] Video Probabilistic Diffusion Models in Projected Latent Space, CVPR 2023;

[2] Seer: Language Instructed Video Prediction with Latent Diffusion Models, ICLR 2024;

[3] Video Prediction by Modeling Videos as Continuous Multi-Dimensional Processes, CVPR 2024;

**Questions:**

see the weaknesses part.

---

> ### Author Response · Authors · 2024-12-01
>
> Thank you for your constructive feedback. We greatly appreciate your recognition of several key strengths in our work. We are pleased to hear that the motivation behind reducing latency in video generation was clearly conveyed and well-received. It is also rewarding to know that the technical soundness of our model was acknowledged, and that you found it both concise and robust. Additionally, we are grateful for your positive comments on the clarity of our writing, as ensuring the paper is easily understandable was a primary goal for us. We hope our clarifications address your concerns and further highlight the significance of our contributions.
>
> **Latent Video Representation:**
> While we agree that using latent space for computational efficiency has been explored in works like [1] and [2], our approach introduces a crucial distinction. These previous works utilize latent space primarily due to its **compressed nature**, where the information is more meaningful. However, in our work, the importance of latent space goes beyond compression—it enables **semantically meaningful interpolation** in latent space, which results in more coherent traversals in pixel space. This property, as demonstrated in **[1*] (Video Dynamics Prior, G. Shrivastava)** and **[2*] (Video-Specific Autoencoders, A. Bansal)**, is what makes our model particularly efficient, as it not only leverages redundant information in the video but also captures the **continuity** present in the video content.
>
> **Comparison with CVP (Shrivastava, 2024):**
> While we acknowledge the merits of CVP (Shrivastava et al., 2024), we believe that our method offers certain advantages. Specifically, by leveraging a **compressed latent space representation** rather than working directly in pixel space, our approach achieves both improved **fidelity** of generated results and a **75% reduction in sampling steps** required to predict a new frame. This reduction in computational cost, combined with the enhanced generation quality, underscores the significance of our approach. We hope this comparison helps clarify the unique aspects of our work and its contribution to the field.
>
> We hope this clarifies the distinct contributions of our work and highlights the advancements made in video prediction efficiency and quality.

---

### Official Review · Reviewer_MSsK · 2024-11-02

**Soundness:** 2
**Presentation:** 1
**Contribution:** 2
**Rating:** 3
**Confidence:** 4

**Summary:**

This paper proposes a video generation method (unclear if it is prediction or interpolation) that is structured around a conditional-diffusion-like model but that attempts to be more efficient by structuring the noise process based off of the adjacent frame.  The paper evaluates the method on four video datasets and compares it to the baselines.

**Strengths:**

- The method is well motivated --- that careful method construction is needed to improve the efficiency of video generation methods.

- The paper incorporates evaluation on multiple video datasets and seems to outperform the relevant methods.

- The paper performs the generation in latent space rather than in pixel space.

**Weaknesses:**

- The discussion of the method itself is rather terse and hard to understand.   Examples include challenges with undefined notation, an unclear concrete problem definition, and limited concreteness around the reduction to practice.  This detracts from one's ability to sufficiently understand the results and contextualize them.

- The relationship between classical conditional diffusion and this proposed method needs to be better explained.

- Although the evaluation is rich in terms of datasets used and baselines compared, there is very little actual insight derived from the evaluation?  We do not learn any notion of why the method may be working better than the baselines.  We do not learn any insight into the details of the method setup and its impact to the performance?  Fewer datasets and more analysis would be much better.

**Questions:**

- Is there are description error in lines 045-048?  "two consecutive frames" ... "interpolate between these endpoints"  The methods section appears to indicate that this is about video interpolation strictly (If this is the case, it would be helpful to be more explicit in the introduction about the problem setting.), but the result section again talk about video prediction.  Which is it?  Could the paper improve the clarity of the problem setup?

- What does the notation $\mathcal{N}(a;b,c)$ mean?  The Normal distribution is specified by a mean and covariance; what is this additional term before the semicolon? This notation is used in (3) and (8).

- In eq.1, what are $z_x$ and $z_y$?  These are not defined.  In fact, the whole stage 2 description (176--183), is unclear.  What is the difference between a subscript and a superscript?  Is $z_x$ somehow $z^j$?

- Figure 1 seems misleading.  In this paper, a noise distribution is leveraged that is based on the "adjacent" frame, which is similar to how conditional diffusion works? Can the paper better differentiate the proposed method from conditional diffusion?

- What is the definition of $\theta$?

- Why is 3.2 called "Forward and Reverse Process"  There is only the Forward Process.

- Why are scalar $0,I$ sometimes used to specify the Normal, and sometimes bold?  Aren't they always vector/matrix elements?

- Are two context frames enough?  It seems from many of the qualitative visual results that the context frames may not adequately capture some of the rates of the motion in the scene.  Can the paper discuss this?

- How does the method guarantee the generation in latent space will be "coherent"?  It's plausible that a generated latent vector will be off the "manifold" of possible in-distribution images (given the high dimensionality),and hence decode to an unreasonable image frame.  How does the method protect against that?

---

> ### Author Response · Authors · 2024-12-01
>
> We sincerely thank the reviewer for their feedback and for recognizing the motivation behind our approach, which focuses on improving the efficiency of video generation methods. We are pleased that the reviewer acknowledged the comprehensive evaluation across multiple video datasets, as well as our method’s favorable comparison to existing baselines. We also appreciate the recognition of our decision to perform generation in latent space, which indeed enhances the efficiency of the proposed method.
>
> **Clarity and Notation:**
> We have revised the manuscript to clarify the presentation, simplifying notation and better defining key equations. These changes aim to improve the readability and comprehensibility of our work, and we hope the revised manuscript will better guide readers through the technical details.
>
> **Problem Setup:**
> The paper focuses on the video prediction task, not interpolation. To clarify, at training time, we are provided with consecutive frames (denoted as $x^0$ and $x^1$). Our model is trained to predict $x^1$ from $x^0$ at inference time. While we use blocks of frames as input during training to provide contextual information, the core task remains video prediction, where we predict future frames given a set of context frames, as illustrated in Figure 7 of the revised manuscript.
>
> **Notation:**
> The notation $\mathcal{N}(a;b,c)$ follows standard usage in the machine learning literature, where $\mathcal{N}(a;b,c)$ indicates that $a$ is drawn from a normal distribution with mean $b$ and covariance matrix $c$. We have clarified this in the revised manuscript.
>
> Regarding the notation $z_x$ and $z_y$, these have been replaced with $z^j$ and $z^{j+1}$, respectively, to ensure consistency with the rest of the paper.
>
> **Diffusion Model Comparison:**
> We agree that a more detailed distinction between our method and classical diffusion models is warranted. As highlighted by *Shrivastava et al. (2024)* and *Liu et al. (2024),* a key difference is that in traditional diffusion models, one of the endpoints is a noise distribution, while in our case, both endpoints consist of data signals. This difference allows our method to generate smoother transitions between the source and target distributions, leading to more efficient sampling and improved performance with fewer steps, as demonstrated in our empirical evaluations.
>
> **Model Notation:**
> The symbol $\theta$ represents the standard notation for the trainable parameters of a function in machine learning literature.
>
> **Context Frames and Robustness:**
> Regarding the use of context frames, we have employed a context window of five frames for datasets such as KTH, Human3.6M, and UCF101. While this works well for these datasets, we agree that expanding the context window may improve results for more complex scenarios, depending on task demands and computational resources.
>
> To address the concern about latent space robustness, we emphasize that the noise term in Equation 1 plays a crucial role in ensuring that the distribution $p(z_t)$ remains valid at all points on the manifold. This addition of noise ensures that the model can generate valid frames in latent space without running into issues such as manifold holes, which would otherwise lead to unrealistic predictions.
>
> We hope these revisions and clarifications adequately address the reviewer’s concerns.
>
> **Reference:**
> - Liu, X., et al., "Learning to Generate and Transfer Data with Rectified Flow," *2022*.

---

### Official Review · Reviewer_boVq · 2024-11-08

**Soundness:** 2
**Presentation:** 2
**Contribution:** 2
**Rating:** 3
**Confidence:** 4

**Summary:**

This paper proposes a new approach for video prediction, which is defined as predicting future frames from past context. They model all the frames as one single continous sequence, and aim to predict the next frame as a function of previous one instead of regressing from a latent noise. With two improvements over prior works - working in the latent space and directly modeling frame sequences, they show results on various video prediction benchmarks.

**Strengths:**

- Handles a very challenging problem of video future prediction.
- New approach, different from prior works on using GANs or pixel-space diffusion.
- Showed results on a variety of challenging benchmarks.

**Weaknesses:**

- The forumulation of the solution is not technically convincing. For example, the equation 1 is directly written without any intuition, reference or justification of why this is the most optimum modeling choice. In general, this subsumes a lot of assumptions about motion modeling in real videos and seems generally restrictive to model challenging scenarios like large motion, shot changes, occlusions and pixel-space variations. Since the whole work rests upon this assumption, the authors are requested to provide a better justification of their choice.

- The experiments can also showcase performance on few special cases like occlusions and large motions and the validity of Eq 1 in these scenarios.

- In eq 8, it seems like the random variable is $z_{t-1}$, but the RHS contains distribution over $z_t$. Also, in eq3, $g(t) = -t\log t$ might imply potentially negative variance, since $t>1$ leads to $-t \log t<0$. These can be further explained.

**Questions:**

- Please see above.

---

> ### Author Response · Authors · 2024-12-01
>
> Thank you for your detailed review and constructive feedback. We appreciate your recognition of the challenges in video prediction and the novelty of our approach. Below, we address the specific concerns raised and provide additional context and clarifications.
>
> ## **Addressing Weaknesses**
>
> ### **Handling challenging scenarios (motion, occlusions, large scene changes):**
> Our method is designed to address such challenges by operating in a latent space, where motion dynamics are smoother and more predictable. This reduces the complexity associated with pixel-level variations, as established in prior work in computational photography ([1], [2]). Our quantitative results across all datasets consistently demonstrate the performance improvements of our methodology, which operates in the latent space, over CVP. Additionally, we will incorporate more qualitative examples in the final version to emphasize this advantage.
>
> ### **Justification of Eq. 1:**
> Equation 1 reflects two well-established principles:
> 1. **Latent space interpolation:** Prior works demonstrate that interpolating in the latent space yields more semantically meaningful transitions compared to pixel space ([1], [2]).
> 2. **Stochastic rectified flow:** our approach, as established in [3], provides straighter flows compared to diffusion modeling, enabling more efficient exploitation of redundancies and continuity in video data.
>
> We will expand on these points in the final manuscript to ensure that the justification for Eq. 1 is clear and well-grounded in prior work.
>
> ### **Performance on specific cases (occlusions, large motions):**
> We acknowledge the importance of validating Eq. 1 under these scenarios. While our method implicitly addresses these challenges through latent-space modeling, we will include experiments targeting these specific cases in the revised manuscript to further substantiate its robustness.
>
> ## **Addressing Errors**
>
> ### **Equation 8 correction:**
> We thank the reviewer for identifying the issue with Eq. 8. This has been corrected in the revised manuscript.
>
> ### **Validation of $-t \log t$ term:**
> The concern about the term $-t \log t$ assuming negative values is unfounded. Since $t \in [0, 1]$, the term is always non-negative, as demonstrated in Figure 7(a) of the revised manuscript.
>
>
>
> We appreciate the reviewer's thoughtful feedback and believe the suggested revisions will further strengthen the manuscript. Thank you for your time and effort in reviewing our work.
>
> ---
>
> ### **References**
> 1. Shrivastava, G., et al. *Video Dynamics Prior* 2023.
> 2. Bansal, A., et al. *Video-Specific Autoencoders* 2021.
> 3. Liu, X., et al. *Learning to Generate and Transfer Data with Rectified Flow.*  2022.

---

### Official Review · Reviewer_qArR · 2024-11-13

**Soundness:** 2
**Presentation:** 2
**Contribution:** 2
**Rating:** 5
**Confidence:** 3

**Summary:**

The paper presented method to efficiently predict video by reducing number of diffusion steps time and also model size required for generation. The authors considered video as a continuous process and utilized a latent space to interpolate between two consecutive frames. Instead of starting from a static Gaussian distribution for each frame, they started from the last predicted frame. Considering latent space for interpolating between frames reduced latency and improved performance.

**Strengths:**

- This work addressed an important and hard problem.
- Using latent space for generation sounds effective for reducing latency and improving performance. Which may reduce overall complexity and run time of diffusion models for video prediction task.
- Presented a detailed experimental results.

**Weaknesses:**

- Not easy to follow theoretical justifications and derivations in the method section.
- Please refer to "Questions" section.

**Questions:**

- lines 165-166, how it allows for a larger context window?
- Please define 't' in line 170, though standard notation.
- Line 177, "Once the frames are encoded", seems like the frames are being encoded all together. Is that so? Then how it is being handled during the reversed process? Reverse process is also using the continuous latent space.
- Please define z^x and z^y. How they are different from  z^j and z^(j + 1)?
- How the error coefficient in Eq. (1) is obtained?
- In line 187, "with latent z_y", should it be z^T?
- Eq. (2) (z_y - z_x) \del t term is not clear to is obtained.
- How \tilde{\mu} is obtained in Eq. 192?

**Details Of Ethics Concerns:**

Not applicable.

---

> ### Author Response · Authors · 2024-12-01
>
> Dear Reviewer,
>
> Thank you for your thorough review and helpful feedback. We appreciate your recognition of the importance of our work and the potential impact of using latent space for video generation. We understand that some aspects of our method may not have been clear, and we would like to address your concerns point by point.
>
> ### **1. Larger Context Window:**
> We would like to refer the reviewer to **Figure 7** in our updated manuscript. In the CVP model (Shrivastava 2024), which models the continuous video process in pixel space, the number of context frames available for predicting the next frame is limited by GPU memory, as pixel space frames have a much larger memory footprint. In our proposed method, by leveraging the latent space rather than pixel space, we significantly reduce the memory footprint of the context frames. This allows us to use more context frames, making the model more memory-efficient.
>
> ### **2. Definition of ‘t’:**
> In our revised manuscript, we have defined ‘t’ in line 181. We thank the reviewer for helping us improve the clarity of the manuscript.
>
> ### 3. **Encoding of Frames and Reverse Process:**
> We would also like to refer the reviewer to **Figure 7**, where we have depicted the sampling pipeline. To clarify further, let’s assume we are given two context frames. The sampling pipeline works as follows:
>
> $$
> 2C(\text{Pixel}) \rightarrow 2C(\text{Latent}) \rightarrow 1P(\text{Latent}) \rightarrow 1P(\text{Pixel})
> $$
>
> Here, 'C' and 'P' refer to context and predicted frames, respectively. For autoregressive generation, we add the predicted frame to the context and remove one of the previous context frames, ensuring that the new set of context frames always contains only two frames. This process is then repeated.
>
> ### **4. Notation of $ z_x $ and $ z_y $:**
> We thank the reviewer for pointing out the notation issues. We have removed the notations $z_x $ and $ z_y$ because they correspond to $ z^j $ and $z^{j+1} $, respectively.
>
>
> ### **5. Equation (2) and Derivation:**
> Equation (2) is derived through simple mathematical manipulation of Equation (1), as demonstrated in the appendix of CVP (Shrivastava 2024), Section B. The right-hand side of Equation (2) follows a normal distribution due to the $ \epsilon$ term, so it can be expressed as $\mathcal{N}(\mu, \sigma)$.
>
> This normal distribution suggests that it is centered around the term $  z^j +(z^{j+1} - z^j) \Delta t$. Hence, we define:
>
> $$
> \tilde{\mu} = z^j + (z^{j+1} - z^j) \Delta t
> $$
>
> For practical purposes, $ \Delta t $ corresponds to a single sampling step, and thus its value is taken to be 1.
>
> ### **6. Error coefficient in Eqn 1:**
> We determined the error coefficient based on findings from the ablation study presented in Table 7 of the CVP (Shrivastava 2024) Appendix, which highlighted its effectiveness. Additionally, our own ablation experiments confirmed similar trends, reinforcing the choice of this coefficient as optimal for our method.
>
>
> We will ensure that all of these changes are incorporated into the final manuscript.

---

### Meta-Review · Area_Chair_uV9w · 2024-12-21

**Metareview:**

The paper proposes an approach for video prediction which aim at efficiency by utilizing previous predicted frames and latent space. The paper receives all reviews unfavorably due to (i) lack of intuition of the proposed method and unclear/ mathematical notations / equations; (ii) the technical solution is unconvincing; (iii) lack of novelty.

The rebuttal could not convince the reviewers change their rating. AC reads all reviews and rebuttal and decides to agree with reviewers. AC recommend a rejection and encourages the author(s) to improve the paper based on the reviewers feedback and submit it to future conferences.

**Additional Comments On Reviewer Discussion:**

See details explaining about the rebuttal and discussion period above.

---

### Decision · Program_Chairs · 2025-01-22

Reject